# A New *SMAD4* Splice Site Variant in a Three-Generation Italian Family with Juvenile Polyposis Syndrome

**DOI:** 10.3390/diagnostics12112684

**Published:** 2022-11-04

**Authors:** Caterina Micolonghi, Maria Piane, Aldo Germani, Soha Sadeghi, Fabio Libi, Camilla Savio, Marco Fabiani, Rita Mancini, Danilo Ranieri, Antonio Pizzuti, Vito Domenico Corleto, Pasquale Parisi, Vincenzo Visco, Giovanni Di Nardo, Simona Petrucci

**Affiliations:** 1Department of Experimental Medicine, Faculty of Medicine and Dentistry, Sapienza University of Rome, 00161 Rome, Italy; 2Department of Clinical and Molecular Medicine, Faculty of Medicine and Psychology, Sapienza University of Rome, 00189 Rome, Italy; 3S. Andrea University Hospital, 00189 Rome, Italy; 4ALTAMEDICA, Human Genetics, 00198 Rome, Italy; 5Medical Genetics Unit, IRCCS Mendel Casa Sollievo della Sofferenza, 71013 San Giovanni Rotondo, Italy; 6Department of Medical-Surgical Sciences and Translational Medicine, Faculty of Medicine and Psychology, Sapienza University of Rome, 00189 Rome, Italy; 7Department of Neuroscience, Mental Health and Sense Organs (NESMOS), Faculty of Medicine and Psychology, Sapienza University of Rome, 00189 Rome, Italy

**Keywords:** juvenile polyposis syndrome, *SMAD4*, splice site variant

## Abstract

Juvenile polyposis syndrome (JPS) is an autosomal dominant disorder characterized by hyperplastic polyps in the upper and lower gastrointestinal (GI) tract with a high risk of developing GI cancers. We have described a three-generation Italian family with all the spectrum of *SMAD4* phenotype. A multigene panel test was performed on the genomic DNA of the proband by next-generation sequencing, including genes related to hereditary GI tumor syndromes. Molecular analysis revealed the presence of the c.1140-2A>G substitution in the *SMAD4* gene, a novel splice variant that has never been described before. Our family is remarkable in that it illustrates the variable expressivity of the *SMAD4* phenotype within the same family. The possibility of phenotype variability should also be considered within family members carrying the same mutation. In JPS, a timely genetic diagnosis allows clinicians to better manage patients and to provide early surveillance and intervention for their asymptomatic mutated relatives in the early decades of life.

## 1. Introduction

Juvenile polyposis syndrome (JPS) is an autosomal dominant disorder characterized by hamartomatous polyps in the upper and lower gastrointestinal (GI) tract with a high risk of developing GI cancers. The age of clinical presentation is highly variable, as JPS can occur in both children and adults, but usually by the age of 20 years. The term “juvenile” refers to the type of polyps, rather than to the age of disease onset. Juvenile polyps are characterized by goblet cells and tubules with columnar epithelium lined on their stroma. These structures vary in size, shape (sessile or pedunculated) and number, ranging from 1 to more than 100 in their lifetime. Often the epithelium of the polyps can become ulcerated leading to the infiltration of inflammatory cells, which is the first step in a series of sequential events. As the juvenile polyp becomes inflamed and enlarged, the glands and crypts begin to fill with mucus and progress to the classic hamartomatous juvenile polyp [1]. Untreated polyps can bleed and cause anemia, rectal bleeding, abdominal pain, and diarrhea. Surveillance is essential to reduce the risk of developing cancer. Indeed, polyps can take on some of the characteristics of tubular or villous adenoma and develop into adenocarcinoma [2]. In families with JPS, the risk of GI cancers (mostly colon but also upper GI and pancreatic cancer) ranges from 9% to 50% [3]. A differential diagnosis should be considered if the presence of polyps is associated with additional features not associated with JPS. There are many syndromes with polyps’ predisposition in differential diagnosis with JPS, including *PTEN* hamartoma tumor syndrome, nevoid basal cell carcinoma syndrome, Peutz-Jeghers syndrome, and Lynch syndrome. According to current research, two genes have been associated with JPS: *SMAD4* (MIM * 600993) and *BMPR1A* (MIM * 601299). In 40–50% of cases, JPS is caused by a disease-causing germline variant in these genes, with a higher likelihood of family history and a greater risk of colectomy [4]. The detection of pathogenic mutations in *BMPR1A* or *SMAD4* genes confirms the clinical diagnosis of JPS, even if the clinical features are inconclusive. Genetic testing is essential to determine the risk to patients’ relatives and to provide them with recommendations for screening. Germline mutations in *SMAD4* and *BMPR1A* disrupt the signal transduction pathway of the transforming growth factor β (TGFβ). The *SMAD4* protein is a mediator of TGFβ signaling pathways. It forms a complex that moves into the nucleus and regulates gene expression. It has two functional domains, MH1 and MH2, which are a DNA binding domain and a transcriptional activator, respectively, involved in the formation of the complexes made by Smad proteins [5]. The BMPR1A protein is a type I cell surface receptor for the BMP pathway. These proteins bind to DNA sequences to regulate transcription and their dysfunction causes unregulated cell growth, that can lead to polyp formation [6,7]. Clinical differences in symptoms between *SMAD4*/*BMPR1A* are still debated. Carriers of the pathogenic *SMAD4* variant more readily develop GI polyps than *BMPR1A* carriers and have a higher risk of gastric cancer [7,8]. *SMAD4* carriers have an increased risk of anemia, hemorrhagic telangiectasia, and a higher prevalence of juvenile gastric polyps, compared to *BMPR1A* mutated cases [9]. In addition, most individuals with a pathogenic *SMAD4* variant present with a combined syndrome of JPS and hereditary hemorrhagic telangiectasia (JPS/HHT). HHT is clinically characterized by epistaxis, visceral arteriovenous malformations (AVM), or mucocutaneous telangiectasias. HHT is suspected when at least two manifestations are present and is diagnosed when there are ≥3 features [8]. The first criteria for the diagnosis of JPS were proposed by Sachatello in 1974; today, JPS is diagnosed when a patient has any of the following criteria: (1) more than five juvenile polyps in the colon or rectum; (2) juvenile polyps in other parts of the gastrointestinal tract; or (3) any number of juvenile polyps and one or more affected family members [10].

## 2. Case Presentation

We have described a three-generation Italian family with exhibiting the full spectrum of *SMAD4* phenotype. The proband, a fifteen-year-old boy (III:6, Figure 1A) was diagnosed with JPS at thirteen years of age. The disease manifested as severe sideropenic anemia requiring multiple transfusions. Colonoscopy revealed multiple pedunculated and sessile hamartomatous polyps in the right colon, sigmoid colon, and rectum, with viable size and shape (Figure 2). Due to high-grade dysplasia detected on the biggest polyp of the right colon, colectomy became necessary. No alterations emerged by esophagogastroduodenal endoscopy. Magnetic resonance angiography of the head and echocardiography were normal. Small multiple juvenile colon polyps were also present in the father (II:3), who underwent gastrectomy for gastric cancer at the age of 37 years, and in the paternal aunt (II:1). Moreover, the uncle (II:2) was diagnosed with Menetrier’s disease (MD) and the grandfather died with a pancreatic adenocarcinoma at age of 33 (I:1). Two cousins (III:4 and III:5), who refer GI disturbances, are awaiting endoscopy. Because of the peculiar phenotypes of the proband and his paternal relatives, genetic counseling was required.

## 3. Materials and Methods

### 3.1. NGS Analysis

After obtaining *ethical approval* and written informed consent, genomic DNA samples of the proband and his relatives were extracted from peripheral blood lymphocytes, according to standard procedures. A multigene panel test, including genes related to hereditary GI tumors syndromes, was performed on genomic DNA of the proband by next-generation sequencing with on Ion PGM Platform (Life Technologies), covering the coding exons and exon-intron boundaries of 25 genes (*APC*, *ATM*, *BARD1*, *BRIP1*, *CDH1*, *CDK4*, *CDKN2A*, *CHEK2*, *EPCAM*, *MLH1*, *MRE11*, *MSH2*, *MSH6*, *MUTYH*, *NBN*, *PALB2*, *PMS2*, *PTEN*, *RAD50*, *RAD51C*, *RAD51D*, *RECQL1*, *SMAD4*, *STK11*, and *TP53* (Appendix A). The identified nucleotides alterations were named conforming to the Human Genome Variation Society nomenclature guidelines (https://varnomen.hgvs.org/ accessed on 27 September 2022). The clinical classification of the variants was carried out according to the American College of Medical Genetics and Genomics (ACMG) criteria. Only damaging mutations and variants of uncertain significance were confirmed by Sanger sequencing.

### 3.2. RNA Analysis

To establish the consequence of the new splicing variant in the *SMAD4* gene, total RNA was extracted by peripheral blood of the proband and family carriers using RNeasy Mini kit (Qiagen, Hilden, Germany) according to the manufacturer’s instructions. RNA quantity and quality were determined by a NanoDrop, and 100–200 ng RNA was converted into cDNA by the iScript cDNA Synthesis Kit (BioRad, Hercules, CA, USA) with Oligo(dT) primers protocol. We designed *SMAD4* forward primer, spanning the junction of exon 6–7 (5′-TACCATCATAACAGCACTACC-3′), and reverse primer, spanning the junction of exon 10–11 (5′-TGACAGACTGATAGCTGGAG-3′), for PCR amplification of the cDNA covering exons 7–11. PCR was performed with AmpliTaq Gold (Thermo Fisher Scientific, Waltham, MA, USA) with the following cycle-program: 96 °C 10 min, (96 °C 30 s, 60 °C 30 s, 72 °C 40 s) × 35, 72 °C 8 min. PCR products were qualitatively assessed in 1% agarose gels, healthy controls were run in parallel with patient samples and were used as a reference. PCR products were purified using PureLink^®^ PCR Purification Kit (ThermoFisher) and bidirectionally sequenced by Sanger on SeqStudio Genetic Analyzer.

### 3.3. Validation of SMAD4 Splice Variant

The validation of the variant identified on the *SMAD4* gene was carried out using a pair of primers (5′-ATTAAGCATGCTATACAATCTGAACTA-3′ and 5′-TGCACTTGGGTAGATCTTATGAA-3′) that allowed us to exclude the amplification of the *SMAD4* pseudogene, which shares almost all exons but not introns with the functional gene.

## 4. Results

The molecular analysis identified the c.1140-2A>G splicing variant in the *SMAD4* gene (Figure 3) and the c.362A>G (p.Tyr121Ser) variant of uncertain significance in the *MSH2* gene, both heterozygous and never described to date (Figure 1B). The two variants were confirmed by Sanger sequencing. Segregation studies on parental DNA showed the maternal origin of the *MSH2* variant and the paternal origin of the *SMAD4* mutation, detected also in the aunt (II:1) and uncle (II:2) (Figure 1A). The analysis of the splice site c.1140-2A>G in the proband and in the mutated relatives by RT-PCR have shown that this variant results in the displacement of the splice acceptor site (r.1140dup) anticipating the splice acceptor site of intron 9 by one base and including the last base of intron 9 in exon 10. This insertion results in a frameshift alteration with the prediction of truncated protein NM_005359.6(SMAD4_i001):p.(Leu381Valfs*12) (Figure 3) and of a spliceogenic loss-of-function effect.

## 5. Discussion and Conclusions

We describe an Italian JPS family in which a new splicing variant in the *SMAD4* gene is segregated with different phenotypes related to the disease (Figure 1B). The missense c.362A>G substitution in the *MSH2* gene, also identified in the proband and his healthy mother, replaces the amino acid tyrosine with serine in position 121, p.(Tyr121Ser). It has never been detected in individuals with hereditary cancer and a functional study demonstrated that it may not impact DNA mismatch repair activity [11]. As the available evidence are insufficient to determine the role of this variant in the etiology of the disease, the c.362A>G in *MSH2* must be classified as a Variant of Uncertain Significance and unusable for clinical purposes. The variant NM_005359.6:c.1140-2A>G in the *SMAD4* gene, is located at the splicing acceptor site of intron 9 predict to result in the alteration of the mRNA splicing process with the production of an absent or reduced protein, by shifting back one nucleotide at the beginning of exon 10 (r.1140dup), with a consequent frameshift and possible production of a truncated protein, p.(Leu381Valfs*12). Splicing variants in the *SMAD4* gene have been associated with Juvenile Polyposis [12,13,14,15]. Recently, a germline splicing site variant of *SMAD4* (c.1139+3A>G) has been described in a 50-year-old woman with a familial history positive for gastrointestinal cancers and multiple gastrointestinal neoformations, but no evidence is reported regarding the consequence of this variant on the splicing process [16]. Mutations with loss of function of the *SMAD4* gene are associated with hereditary hemorrhagic telangiectasia and Juvenile Polyposis Syndrome, with autosomal dominant inheritance [8,13,17]. In this work, the transcriptional study performed on blood-derived mRNA samples of the proband, and familial carriers showed that the nucleotide substitution within the consensus splice site of *SMAD4* intron 9 creates an alternative splice acceptor site, predicted to cause the shift of the reading frame and the production of a truncated protein. Based on the ACMG criteria (PVS1, PM2) the c.1140-2A>G variant identified in this family is classifiable as likely pathogenetic [18]. Phenotypic variability has been described between and within several *SMAD4* families [8]. Even if all GI clinical features were described in the literature in *SMAD4* carriers, until now they never occurred together in relatives of one unique family. In carriers of *SMAD4* pathogenic variants, juvenile polyps may develop in the colon as well as throughout the GI tract [19,20]. Juvenile polyps develop from infancy through adulthood. Most JPS individuals have polyps by age 20 years. They can be located throughout the gastrointestinal tract (27%), in the colorectum alone (36% of case) or in the stomach alone (36% of cases). Malignant tumors develop in 15% of patients. The major risk of cancer development was dependent on the type of polyp distribution. Colorectal cancer is the most frequent. Indeed, the incidence of this type of malignancy reaches is 17%–22% by age 35 years and 68% by age 60 years. The gastric cancer risk is increased in patients with gastric polyps (21% of cases). Less frequently, malignant tumors in pancreas, small intestine, breast, and thyroid may occur [21]. Exceptionally, cases with MD have been described [22]. Some studies hypothesized that also JPS and MD are a manifestation of the same molecular defect. If in JPS the gastric involvement is massive, MD can be misdiagnosed [23]. Although pathogenic mutations in *SMAD4* can cause JPS and JPS/HHT syndrome, in our family no clinical evidence of hereditary hemorrhagic telangiectasia has been referred or identified. A possible genotype–phenotype correlation has been proposed since mutations in the MH2 domain seemed to recur in JPS–HHT patients while other mutations throughout the *SMAD4* gene were found in JPS patients. Currently, such a correlation is still being discussed. Hemorrhagic telangiectasia signs in JPS may be asymptomatic/paucisymptomatic and often underdiagnosed. Moreover, they can be absent at the time of diagnosis and afterward should be deeply investigated in all *SMAD4* damaging mutation carriers [24]. For these reasons, patients should be managed clinically with HHT risk even if without symptoms. Furthermore, in at-risk relatives of the probands, HHT surveillance should begin in childhood, even before the surveillance for polyps, because of the high HHT morbidity and mortality at a young age. Our patient’s little sister (III:7) is still waiting for the segregation test because, considering her young age (4 years old), their parents preferred to postpone the analysis. Genetic test information must be managed carefully because of the psychological impact of the test and the weight of this kind of information both in children and their parents. If not well comprehended, this kind of information can create disastrous social, emotional, psychosocial, and educational consequences in minors and, generally in their family. Nevertheless, a genetic test should be considered whenever a real benefit for a minor exists. Genetic testing is an essential information starting the correct surveillance protocol or precautionary surgical intervention. The concern of parents about tests in minors can delay the divulgation of this kind of information but not the surveillance. On the other hand, surveillance (especially endoscopy) can be disturbing for children. The identification of the familial pathogenic mutation can select those young people who really need such invasive clinical procedures [13].

In conclusion, we describe a new splice site mutation in *SMAD4* gene in a single Italian family where all the carriers manifested different GI phenotypes associated with this gene. Although the HHT recurs in *SMAD4* patients, none in the described family has had clinical features related to HHT. Unfortunately, we could not study each family member in the same deep way. Indeed, only the proband underwent exams aimed to exclude HHT features, while his mutated relatives, asymptomatic for HHT, were investigated only for GI diseases. Moreover, in our family, asymptomatic and paucisymptomatic at-risk relatives that are still waiting for genetic analysis are not even following a clinical management. Generally, *SMAD4* germline mutations carriers should undergo frequent esophagogastroduodenoscopy and colonoscopy (since 18 and 12 years, respectively, if asymptomatic), with a cadence based on the opinion of the reference gastroenterologist and should be screened for signs and symptoms of HHT, even if there is no clinical evidence, because of the high risk of HHT’s complications.

We are aware that the lack of possibility to study all the members of the family may be a limit of our study, as other phenotypic features related to the identified variant could be still undiagnosed. Thus, the identification of other patients with this variant will help to identify the real phenotype associated with this variant.

## Figures and Tables

**Figure 1 diagnostics-12-02684-f001:**
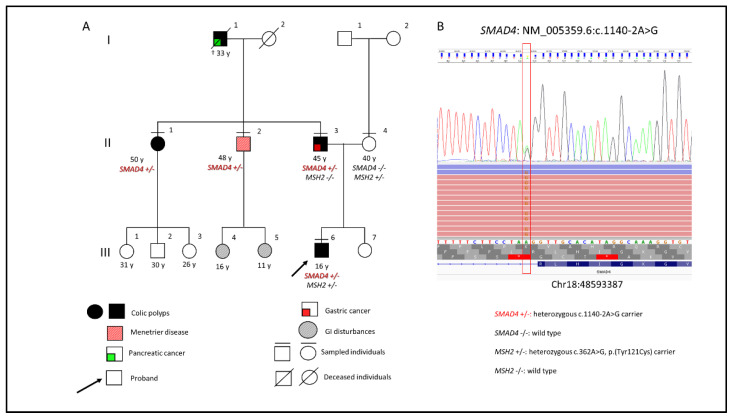
(**A**) Pedigree of the family showing phenotypes of affected relatives and genotypes of sampled individuals. Male is represented by a square; female is represented by a circle. GI, gastrointestinal. (**B**) Multi-gene panel analysis. NGS and Sanger sequencing of the proband showing the intronic heterozygous c.1140-2A>G substitution in *SMAD4* gene on genomic DNA. The identified pathogenic variant in the *SMAD4* gene is visualized by Integrative Genome Viewer (IGV) software. Ref Seq (Reference sequencing) used for variants annotation: NM_005359.6.

**Figure 2 diagnostics-12-02684-f002:**
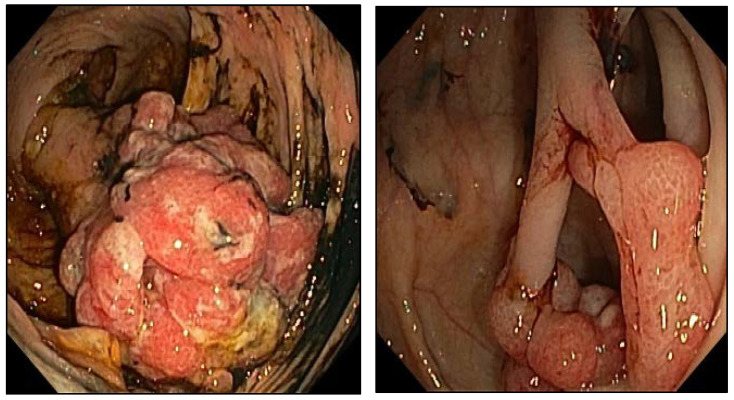
Colonoscopy of III:6 showing a giant cecal polyp (**left**) and a bifid sigmoid polyp (**right**). At histology, lesions were confirmed to be “juvenile polyps”. Dysplastic foci were found in the giant neoformation.

**Figure 3 diagnostics-12-02684-f003:**
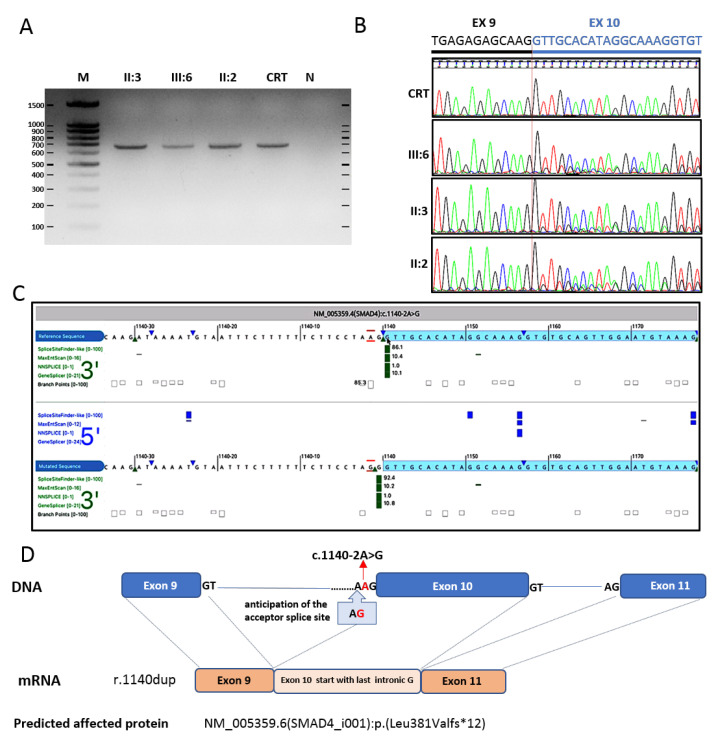
Analysis of the NM_005359.5:c.1140-2A>G acceptor splice variant in ***SMAD4*** gene (**A**) RT-PCR products amplified with primers spanning the junction of exons 6–7 and 10–11 in RNA from leukocytes and separated on 1% agarose gel. M: 100-bp DNA ladder, CRT: normal control, N: PCR control. (**B**) Sanger sequencing of PCR product, indicating displacement of the splice acceptor site and inclusion of the last base of intron 9 in exon 10 in the samples II:2, II:3; III:6 (**C**) Alamut prediction of the splice site variant (Alamut Visual software, version 2.11): the A>G substitution results in the anticipation of the “AG” splice acceptor site of intron 9 by one base and in the inclusion of the last base of intron 9 in exon 10 (green triangle). (**D**) Schematic representation of the c.1140-2A>G variant on genomic DNA (the replacement of the penultimate adenine before the start of the exon 10 with a guanine, in the cerulean box), its effect on the SMAD4 mRNA sequence (anticipation of the start of exon 10 by one nucleotide) and the predicted affected protein (production of a truncated protein).

## Data Availability

All data will be available upon reasonable request to the corresponding author.

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
