# Peer review of "A New SMAD4 Splice Site Variant in a Three-Generation Italian Family with Juvenile Polyposis Syndrome"

_diagnostics, 2022, doi:10.3390/diagnostics12112684_

Round 1

Reviewer 1 Report

Overall:

In this article, the authors reported an Italian juvenile polyposis syndrome (JPS) family in which a new splicing variant in the SMAD4 gene is segregated with different phenotypes. Moreover, they confirmed that this variant results in the alteration of the 171 mRNA splicing process with the production of an absent or reduced protein.

Comments:

1.    Please describe and show the gastric features and the histopathological figure of colon polyp with high-grade dysplasia in a fifteen-year-old boy.

2.    It is difficult to understand the results of figure3c and d. Please explain clearly.

3.    Abstract: Please check “a novel slice variant”.

4.    Page2, line4: Please check “popyp”.

Author Response

We thank you the Reviewer for the suggestions. Here, our point-by-point responses.

  1. Please describe and show the gastric features and the histopathological figure of colon polyp with high-grade dysplasia in a fifteen-year-old boy.

  • We agree with this suggestion. No gastric and duodenal alterations emerged in the proband by the three EGDSs prerformed. We added this information in the text. As the negative results, we decided to not insert the EGDS images in the manuscript. However, if the reviewr deems it necessary, we could send them to him/her or add them in supplementary.

  • The surgery and the histopathologic exams of the proband were performed in another hospital. In order to obtain the figure of the colon polyp and answer the reviewer’s request, we tried to contact the collegues of the other centre. However, they couldn’t be able to help us. Also the relatives of the probands attended the same centre and, for the same reason, their documetation is not available.
  1. It is difficult to understand the results of figure3c and d. Please explain clearly.

We thanks the Reviewer for this suggestion. We modified the legend of figure 3 (points C and D) and described the images in a more detailed way, as the following:

C) Alamut prediction of the splice site variant (Alamut Visual software, version 2.11): the A>G substitution results in the anticipation of the “AG” splice acceptor site of intron 9 by one base and in the inclusion of the last base of intron 9 in exon 10 (green triangle). D) Schematic representation of the c.1140-2A>G variant on genomic DNA (the replacement of the penultimate adenine before the start of the exon 10 with a guanine, in the cerulean box), its effect on the SMAD4 mRNA sequence (anticipation of the start of exon 10 by one nucleotide) and the predicted affected protein (production of a truncated protein)”.

We also added a box with the nucleotidic substitution detailing the anticipation of the “AG” splice acceptor site in figure 3D.

3. Abstract: Please check “a novel slice variant”.

Thank you for this alert. We provided to correct the typo.

  1. Page2, line4: Please check “popyp”.

Thank you for this alert. We provided to correct the typo.

Reviewer 2 Report

The authors described a three-generation Italian family with all the spectrum of SMAD4 phenotype. A multigene panel test was performed on the genomic DNA of the proband by next generation sequencing, including genes related to hereditary GI tumor syndromes. Molecular anayysis revealed the presence of the c.1140-2A>G substitution in the SMAD4 gene, a novel slice variant that has never been described before.

The manuscript was generally well written. My questions and comments to the authors are as follows:

1.     Abstract, introduction: Juvenile polyposis syndrome (JPS) is characterized by “juvenile” (or “hamartomatous”) polyps, not hyperplastic polyps.

2.     Page 3: The figure legend of figure 2 should be next to figure 2.

3.     Figure 2: We cannot see dysplastic foci in endoscopic pictures. Can you provide the microscopic picture of the juvenile polyp ?

4.     Discussion, line 216: What does this sentence ”it risk-relatives of the probands” mean ?

Besides, there are some spelling errors:

1.     Introduction, line 49: The word “popyp” should be spelled as “polyp”.

2.     Case presentation, line 85: The word “sigma” should be spelled as “sigmoid colon”.

3.     Discussion, line 207: The abbreviation “HTT” should be spelled as “HHT”. 

Author Response

We thank you the Reviewer for the suggestions. Here, our point-by-point responses.

  1. Abstract, introduction: Juvenile polyposis syndrome (JPS) is characterized by “juvenile” (or “hamartomatous”) polyps, not hyperplastic polyps.

We than you the reviewer for this correction. We provided to modify it in the text.

  1. Page 3: The figure legend of figure 2 should be next to figure 2.

We provided to move the legend after the figure 2

  1. Figure 2: We cannot see dysplastic foci in endoscopic pictures. Can you provide the microscopic picture of the juvenile polyp ?
  • We agree with the Reviewer about the importance of the microscopic picture of the polyp. The surgery and the histopathologic exams of the proband were performed in another hospital. We tried to contact the collegues of the other centre, in order to obtain the images. However, they couldn’t be able to help us. Currently, the documetation is not available.

  1. Discussion, line 216: What does this sentence ”it risk-relatives of the probands” mean ?

We apologize for this typo. The sentence has been modified as “Furthermore, in at risk-relatives of the probands, HHT surveillance should begin in childhood,”.

Besides, there are some spelling errors:

  1. Introduction, line 49: The word “popyp” should be spelled as “polyp”.
  2. Case presentation, line 85: The word “sigma” should be spelled as “sigmoid colon”.
  3. Discussion, line 207: The abbreviation “HTT” should be spelled as “HHT”. 

We provided to correct all the typos identified by the Reviewer.

Reviewer 3 Report

It is an interesting paper about a new SMAD4 splice site variant in a family with juvenile polyposis syndrome (JPS). Even if multiple germline variants of SMAD4 have been described in the literature, the originality of the case consists of the association of different phenotypes of the same mutation in a single family.

I have some suggestions:

1.      Discussions should be focused on the results obtained, not on generalities about JPS (e.g. 196-203); these can be presented in the introduction.

2.      The same observations are also valid for conclusions (they are too general, they should be reformulated and adapted to the presented case).

3.      It would be helpful to add the strengths and limitations of the work.

At the same time, there are some minor errors:

·        In the introduction, there are not enough references (rows 33-43);  

·        Row 47: gastric cancer is included in upper GI tract cancers;

·        Row 49 polyp instead of popyp;

·        Row 76 “; “instead of “,”

·        Row 143 figure 3 instead of fig 2

·        204-206 Menetrier’s disease was already abbreviated ( MD);

·        210, 211 JPS instead of JP. 

Author Response

We thank you the Reviewer for the suggestions. Here, our point-by-point responses.

  1. Discussions should be focused on the results obtained, not on generalities about JPS (e.g. 196-203); these can be presented in the introduction.

We thanks the Reviewer for this suggestion. We modified this part deleting  the sentences about the JPS genetralities, already detailed in the introduction.

  1. The same observations are also valid for conclusions (they are too general, they should be reformulated and adapted to the presented case). AND

  1. It would be helpful to add the strengths and limitations of the work.

Thank you for these two observations. We reformulated the conclusions, trying to adapt them to the presented case, and we added strengths and limitations of the work, as the following:

“In conclusion we describe a new splice site mutation in SMAD4 gene in a single Italian family where all the carriers manifested different GI phenotypes associated with this gene. Although the HHT recurs in SMAD4 patients, none in the described family has had clinical features related to HHT. Unfortunately, we could not study each family member in the same deep way. Indeed, only the proband underwent exams aimed to exclude HHT features, while his mutated relatives, asymptomatic for HHT, were investigated only for GI diseases. Moreover, in our family asymptomatic and paucisymptomatic at risk-relatives that are still waiting for genetic analysis, are not even following a clinical management. Generally, SMAD4 germline mutations carriers should undergo frequent  esophagogastroduodenoscopy and colonoscopy (since 18 and 12 years, respectively, if asymptomatic), with a cadence based on the opinion of the reference gastroenterologist and should be screened for signs and symptoms of HHT, even if there aren’t clinical evidence, because of the high risk of HHT’s complications. We are aware that the lack of possibility to study all the members of the family may be a limit of our study, as other phenotypic features related to the identified variant could be still undiagnosed. Thus, the identification of other patients with this variant will help to identify the real phenotype associated with this variant. “

At the same time, there are some minor errors:

  • In the introduction, there are not enough references (rows 33-43);  
  • Row 47: gastric cancer is included in upper GI tract cancers;
  • Row 49 polyp instead of popyp;
  • Row 76 “; “instead of “,”
  • Row 143 figure 3 instead of fig 2
  • 204-206 Menetrier’s disease was already abbreviated ( MD);
  • 210, 211 JPS instead of JP. 

We provided to correct all the typos identified by the Reviewer and we provided to add the missing reference.